# Tryptophan: A Unique Role in the Critically Ill

**DOI:** 10.3390/ijms222111714

**Published:** 2021-10-28

**Authors:** Marcela Kanova, Pavel Kohout

**Affiliations:** 1Department of Anaesthesiology and Intensive Care Medicine, University Hospital Ostrava, 708 52 Ostrava, Czech Republic; 2Institute of Physiology and Pathophysiology, Faculty of Medicine, University of Ostrava, 703 00 Ostrava, Czech Republic; 3Department of Internal Medicine, 3rd Faculty of Medicine, Charles University Prague and Teaching Thomayer Hospital, 140 59 Prague, Czech Republic

**Keywords:** delirium, inflammation, kynurenine pathway, polyneuromyopathy, post intensive care syndrome, rapamycin, sepsis, serotonin pathway, tryptophan metabolism

## Abstract

Tryptophan is an essential amino acid whose metabolites play key roles in diverse physiological processes. Due to low reserves in the body, especially under various catabolic conditions, tryptophan deficiency manifests itself rapidly, and both the serotonin and kynurenine pathways of metabolism are clinically significant in critically ill patients. In this review, we highlight these pathways as sources of serotonin and melatonin, which then regulate neurotransmission, influence circadian rhythm, cognitive functions, and the development of delirium. Kynurenines serve important signaling functions in inter-organ communication and modulate endogenous inflammation. Increased plasma kynurenine levels and kynurenine-tryptophan ratios are early indicators for the development of sepsis. They also influence the regulation of skeletal muscle mass and thereby the development of polyneuromyopathy in critically ill patients. The modulation of tryptophan metabolism could help prevent and treat age-related disease with low grade chronic inflammation as well as post intensive care syndrome in all its varied manifestations: cognitive decline (including delirium or dementia), physical impairment (catabolism, protein breakdown, loss of muscle mass and tone), and mental impairment (depression, anxiety or post-traumatic stress disorder).

## 1. Introduction

Tryptophan (Trp) is the largest of the three aromatic amino acids, with a benzoic nucleus and a pyrrole ring on the side chain. With the molecular formula of C_11_H_12_N_2_O_2_, it is categorized as a glucogenic/ketogenic amino acid. It is also an essential amino acid since the human body lacks the enzymatic machinery for Trp synthesis. Of the eight essential amino acids, Trp has the lowest reserves in the body, and thus Trp deficiency rapidly manifests itself under various catabolic conditions. Hence, it is critical to consume Trp in sufficient quantities through the diet—rich sources include dairy products, eggs, meat, fish, and chocolate. Following its release from dietary protein, Trp passes through the intestinal epithelium into mesenteric and then into systemic circulation. Intestinal microbiota metabolizes Trp to indoles and thus delivers another important pool of tryptophan metabolites into systemic circulation.

Trp and its metabolites play critical roles in a number of physiological processes. They influence immune reactions, have antioxidant properties, and can function as anabolic signals. Apart from this, they act as regulators of the circadian rhythm, and can thus modulate sleep disorders or cognitive functions. Given these diverse roles, Trp metabolism has a significant impact on the clinical outcome of critically ill patients. 

Trp is metabolized through two pathways that include a series of active metabolic intermediates themselves capable of performing signaling functions (Figure 1).

The aromatic Trp core is conserved in the ***serotonin pathway***. The first step in the metabolic cascade is Trp hydroxylation to form 5-hydroxytryptophan (5-HTP, oxitriptan). This step is catalyzed by tryptophan hydroxylase (TPH), and of its two isoenzymes, TPH1 is active in peripheral tissues (gastrointestinal tract, skin) while TPH2 is active in neuronal cell types. Subsequently, aromatic acid decarboxylase (AAAD) removes the carboxyl group to form 5-hydroxytryptamine (5-HT, serotonin). The final product of this pathway is melatonin (N-acetyl-5-methoxy-tryptamine).

The majority (up to 90%) of all Trp is metabolized by the ***kynurenine pathway***, where the intermediates are ligands for Aryl hydrocarbon receptors (AHRs) that primarily influence immune responses. In this metabolic pathway, the aromatic core is destroyed. The limiting step for its initiation is activation of two oxidoreductases—indolamine-2,3-dioxygenase (IDO) and tryptophan-2,3-dioxygenase (TDO). The IDO1 isoform is expressed in most tissues at low levels, including in macrophages, placenta, and epididymis, but not in the liver. The IDO2 isoform is found in the liver, brain, testes, endometrium, placenta, thyroid and can be induced in antigen- presenting cells and B lymphocytes. TDO is found mainly in the liver, but can also be expressed in bone marrow, the immune system, muscle tissue, the urinary and gastrointestinal tracts, and in the brain. Both IDO isoforms (IDO1, IDO2) are activated by proinflammatory cytokines, the most potent IDO1 activator being interferon γ. On the other hand, TDO is mainly activated by cortisol and the hypothalamic-pituitary-adrenal axis (HPA). The catabolites referred to as kynurenines include kynurenine (Kyn), kynurenic acid (KynA), quinolinic acid (QA) and the terminal products niacin (vitamin B3) and nicotinamide adenine dinucleotide (NAD+). The first rate-limiting step in the kynurenine pathway is conversion of Trp to N-formylkynurenine. This step is mediated by TDO in the liver. Under normal conditions, this is the major metabolic pathway. But expression levels of TDO are regulated by systemic levels of Trp, determined mainly by food intake and corticosteroids. Elevated levels of circulating glucocorticoids due to stress, sleep deprivation, or lack of physical activity, stimulate TDO expression. This leads to enhanced liver kynurenine metabolism and the subsequent increased delivery of kynurenines into the CNS. The situation is different with chronically high cortisol levels—it leads to glucocorticoid resistance, suppresses liver TDO expression and shifts to extrahepatic kynurenine metabolism.

IDO can be activated during inflammation (stimulated by TNF, interferon, etc.) to form Kyn, thus diminishing the amount of Trp available for the formation of serotonin, melatonin, and other important azoles (leading to impairment of sleep and cognitive function in critically ill patients). When the kynurenine pathway is fully activated (e.g., in sepsis with a typical increase in the Kyn/Trp ratio), it leads to Trp depletion, the consequent decreased activation of the rapamycin system and the attenuation of proteosynthesis. In critically ill patients, this might result in rapid progression of muscle atrophy, sarcopenia, and polyneuromyopathy [1,2,3].

In this review, we focus on Trp and its metabolites from the perspective of the critically ill patient. Some of the metabolites are involved in anabolic processes, while a larger portion are metabolized to kynurenines, which play an important role as mediators in inter-organ communication, thereby influencing a wide variety of processes. From the point of view of the critically ill patient, the most important is their role in the maintenance of skeletal muscle mass. The neuroexcitatory effects of kynurenines and the products of the other metabolic pathway—serotonin and melatonin—are involved in influencing cognitive function, circadian rhythm, and the development of depression and delirium. We discuss the molecular mechanisms in how activation of Trp metabolism could be involved in controlling inflammation, where we describe the local effects of the IDO pathway. We address its possible role in sepsis, still the most common cause of death in critically ill patients. The modulation of Trp–Kyn metabolism using diet, exercise, and pharmacological intervention could help prevent post-intensive care syndrome (cognitive decline, polyneuromyopathy with both physical and cognitive impairment) and positively influence morbidity and mortality in critically ill patients.

## 2. The Role of Tryptophan in Sepsis and Inflammation

Inflammation is a vital response of the body to all forms of tissue damage caused variously by mechanical trauma, surgery, ischemic or thermal injury, or as a result of infection. The immune response initially remains local, but when there is severe tissue damage, it can degenerate into the so-called systemic inflammatory response syndrome (SIRS) or sepsis following an infection. The role of the inflammatory response is to restrict the area of damage, to stop the infection from spreading, and to repair damaged tissue and thus help restore tissue homeostasis. Since an out-of-control immune response can be lethal, its operation must be effectively coordinated. Sepsis, defined as life-threatening organ dysfunction caused by an unregulated host response to infection, is still one of the most common causes of death in critically ill patients [4]. “Metabolic immune regulation” modulates the immune response through local nutrient supply and Trp has a unique role to play in the modulation of this kind of homeostasis [5].

Immune cells recruited to an inflammatory lesion are exposed to many environmental stresses that trigger cellular signaling pathways. Similar to hypoxia, tissue inflammation activates hypoxia-inducible factor 1α, which modifies tissue metabolism to impaired oxygen supply by increasing the expression of genes responsible for glycolysis. Inflammation due to pro-inflammatory cytokines induces the expression of IDO—and thereby the Trp catabolic Kyn pathway—in antigen presenting cells (APCs: dendritic cells, macrophages, B lymphocytes), and in epithelial cells, vascular endothelial cells, and tumor cells. The simultaneous depletion of Trp and the production of Kyn modulates the immune response. IDO expression is tightly regulated by exogenous signals, and in the absence of inflammation plays only a minor role in Trp degradation. At the mRNA level, IDO transcription is promoted by forkhead box protein 3 (Foxo3) and interferon regulatory factor 8 (IRF-8). IDO is an intracellular enzyme and is not secreted, but although its metabolic effect is initially local, it is not only restricted to IDO-expressing cells. Both Trp deficiency and excess Kyn can be sensed by neighboring cells. As the tissue storage of Trp is low, it can be rapidly depleted by activated IDO, and patients with sepsis can thus be put at risk of inflammation [6] (Figure 2).

Trp levels, which affect two important nutrient sensing systems, play a critical role in the metabolic regulation of inflammation. On the one hand, Trp activates general control non-depressible 2 stress kinase (GCN2) and on the other hand, it inhibits the mechanistic target of rapamycin complex1 (mTORC1). Depletion of Trp (either due to amino acid deficiency or imbalance) activates GCN2, leading to phosphorylation of its downstream target eukaryotic initiation factor 2α (eIF2α). In APCs, this is followed by the production of anti-inflammatory cytokines such as IL10 and the recruitment of regulatory T cells (Treg). Kynurenine can inhibit cytotoxicity of T cells and natural killer cells. This signaling role of IDO/GCN2 is not limited to immune cells but is involved in limiting inflammatory tissue damage, e.g., by inducing autophagy. Thus, activation of Trp metabolism has an anti-inflammatory and immunosuppressive effect, restoring homeostasis in the organism. On the other hand, Trp depletion blocks mTORC1 (a central regulator of cellular functions), which has a crucial role in proteosynthesis and in regulating skeletal muscle mass. The signaling role of IDO/mTORC1 is also important for immune cell activation [7,8].

## 3. Metabolic Control by IDO of the Immune Response in Viral and Bacterial Infections

Increased IDO activity is seen in a number of infectious diseases—it plays a role in viral infections (e.g., *HIV*, *HCV*, *EBV*, *influenza*, *SARS-CoV-2*, and others), as well as in several bacterial infections (*mycobacteria*, *chlamydia*, *listeria*, and others). Inflammation-related IDO activity is often measured by the Kyn/Trp ratio in blood. A high Kyn/Trp ratio is an indicator of the degree of activation of this metabolic pathway controlling the immune response. As a rule of thumb, high Kyn levels are found in patients with community acquired pneumonia (CAP), acute respiratory distress syndrome (ARDS), and in patients who develop multiorgan dysfunction syndrome (MODS) following polytrauma with bacterial sepsis [9].

Activation of IDO by interferon IFNα, IFNγ, TNF, and cytokines in APCs leads to a rapid suppression of viral and bacterial replication. Many of these pathogens are dependent on sufficient Trp being available in the microenvironment (i.e., they are Trp auxotrophs). Degradation of Trp by IDO-expressing cells of the innate immune system is considered the fundamental mechanism of defense against infection. In addition to Trp depletion, Trp metabolites, such as Kyn, have an antibacterial effect, directly suppressing pathogen replication (e.g., *toxoplasmosis*, *chlamydial infection*) or limiting the spreading of viral infection [6]. This antiproliferative effect of IDO prevents the spread of viruses, bacteria, protozoa, as well as tumor cells. On the other hand, IDO activation attenuates immune reactions in host cells, and can thus aid the progress of infection. Metabolic immune regulation essentially serves to protect us from hyperreactivity of our own immune system, from a cytokine storm, enabling the induction of immune tolerance. Our recent experience with *SARS-Cov2* has revealed that the severity of the disease (that then leads to ARDS, and the development of MODS) depends on the extent of cytokine secretion triggering a cytokine storm—and so medications other than corticosteroids that modulate the immune response have not been significantly effective. A high Kyn/Trp ratio can therefore be used as an indicator of the activation of first-line innate immune defense and mortality in critically ill patients [9,10].

IDO is thus a “double-edged sword”, and it can be difficult in therapy to achieve a balance between its antipathogenic effects and defending the individual against overwhelming immune response.

### 3.1. The Role of IDO in Viral Infections 

IDO catalyzes the first and limiting step of Trp catabolism. The main activators in antigen-presenting cells (APCs), i.e., dendritic cells (DC), macrophages or B cells are IFNγ and IFNα, and this points to a pivotal role for IDO in numerous viral infections. 

We find significant expression of IDO in sites where virus replication occurs, i.e., in dendritic cells, alveolar macrophage and epithelial cells in lung tissue and lung-associated lymphatic nodes, following an influenza infection. IFN mediated IDO induction can establish immune tolerance by Trp depletion and production of Kyn (see Figure 3).

Upon infection with HIV, which leads to severe impairment of the T-cell immune response, maximal levels of IDO are to be seen in the spleen and gut-associated lymphoid tissue. IDO is activated by Tat protein (transactivator regulatory protein) that then activates intracellular signaling via a kinase cascade (Janus kinase JAK, phosphoinositide 3-kinase PI3K), which culminates in the breakdown of Trp into Kyn. The second pathway of IDO activation in APCs is mediated directly by regulatory T cells expressing CTLA-4 (cytotoxic T-lymphocyte antigen 4) on their surface, which then interact with the B7 receptors of APCs. Activation of IDO, and the consequent elevation of Kyn, results in the suppression of T cell function. Both CD4+ effector T cells and CD8+ memory T cells failed to proliferate. Although they express CD64, CD38 markers, antiviral defense is reduced. IDO expression is likely to support immunological suppression in favor of viral replication. Therapeutically, inhibitors of JAKs and PI3K are used to reduce viral replication by blocking IDO activity. Blocking CTLA-4 (Treg) does not appear to be effective [12,13]. In influenza infections, using 1-MT (1-methyl-tryptophan) to inhibit IDO increased CD8+ cell numbers and improved CD4+ cell function in mice. This was reflected in improved recovery of the affected lungs. Moreover, inhibition of IDO by increasing CD8+ might improve the efficacy of the flu vaccine [14].

Increased IDO expression in the liver in *chronic hepatitis B virus (HBV)*, *hepatitis C virus (HCV)* infections contributes to immunotolerance and the chronic course of infection, despite the appropriate T cell response during the acute phase. High levels of Kyn correlate with the degree of liver fibrosis [13]. In *Epstein–Barr virus* infections, activation of IDO by B cells suppresses the expression of member D of the activating receptor NK group 2 (NKG2D) on the surface of NK cells, and the subsequent proliferation of CD4+ and CD8+ cells, thus facilitating virus escape [15].

IDO is considered the major mediator of IFNγ-induced antiviral responses. Trp depletion and Kyn production reduces viral replication. However, subsequent induction of Treg and suppression of T-cell proliferation block the anti-viral response. IDO is involved in the establishment of a microenvironment that enables immune tolerance. Thus, the use of IDO inhibitors (PI3K kinase blockers, JAKs) appears beneficial for therapeutic intervention [12,13].

### 3.2. Role of IDO in Bacterial Infections and Secondary Bacterial Infections

IDO plays a central role in the immune responses to bacterial infection as well. Pathogen-derived TLR ligands (lipopolysaccharide (LPS) from Gram-negative and peptidoglycan (Pg) from Gram-positive bacteria) trigger release of immunostimulatory cytokines (TNFα, IL6), and these cytokines, as also oxidative/metabolic stress, can induce IDO expression in certain APCs. Activation of IDO activity in bacteria-infected cells induces a potent bactericidal microenvironment. Availability of Trp decreases, which causes impairment of intracellular trafficking, low transcription activity and they lose infectivity in Trp auxotrophic bacteria. Kynurenine and its other metabolite, 3 hydroxykynurenine (3HK), have a bactericidal effect and thus also contribute to defense against infection.

The anti-infection defense in the initial phase is subsequently replaced by immune tolerance. Depletion of Trp in the local microenvironment promotes anergy in responding T cells, and the production of Kyn/other Trp catabolites enables the recruitment of T reg cells and potentiates the immune response suppression. This suppression protects the organism against secondary immune disability caused by a massive secretion of pro-inflammatory cytokines—a cytokine storm. It thus acts as a negative feedback loop to terminate inflammatory responses, inducing resistance to pathogens, and is involved in bacterial tolerance (enhances anti-inflammatory IL10 response). On the other hand, the immunosuppressive effect of IDO expression may increase the incidence of bacterial superinfections or secondary infections. A typical example is during airway infection with *influenza virus*, which induces local IDO expression. This then alters the inflammatory response and facilitates the outgrowth of pneumococci leading to secondary bacterial pneumonia [16]. The typical co-infecting bacterial species in e.g., *SARS-Cov2* are *Haemophilus influenzae*, *Staphylococcus aureus*, *Klebsiella pneumoniae*, and *Pseudomonas aeruginosa* as well as the mycotic pathogen *Aspergillus fumigatus* [17,18].

### 3.3. Sepsis and Septic Shock

According to the current definition, sepsis is a life-threatening organ dysfunction caused by an unregulated body response to infection. Occurring in 1–2% of all hospitalized patients, it remains one of the leading causes of death worldwide. Sepsis is encountered in almost a quarter of critically ill patients (25% incidence in ICU patients). Although we know the risk factors and pathophysiology, as well as several therapeutic approaches, the mortality rate from sepsis continues to be stubbornly high (30–50% [4]).

This unregulated immune response is associated with endotoxin shock, and IDO plays an important role here. IDO expression is part of the host defense mechanism against pathogen and responds to pro-inflammatory cytokines. It is sensitive to the presence of LPS from Gram-negative and Pg from Gram-positive bacteria, which—upon recognition of Toll-like receptors (TLR4, TLR2) on the surface of APCs—initiate intracellular signal transduction. The whole pro-inflammatory cascade is activated when NFκB (nuclear factor κB) is transported into the nucleus, where it induces the transcription of genes encoding pro-inflammatory cytokines [18]. The activation of this pro-inflammatory pathway does not discriminate between PAMPs (pathogen-associated molecular patterns) from infectious agents and DAMPs (damage-associated molecular patterns), which are released during burns, polytrauma or following surgery. Therefore, Trp catabolism due to increased IDO expression is common in a heterogeneous group of critically ill patients.

IDO expression is induced in DCs and monocytes in response to IL1, IL6, TNF, IL12 (which prime naive CD4 T cells towards a Th1 phenotype). In contrast, IDO expression is decreased by IL10, which, when produced by DCs, promotes the development of a Th2 phenotype in naive CD4 T cells. IDO activation could have a beneficial as well as a detrimental effect. For instance, the detrimental immune response in endotoxin shock may also occur through IDO modulation [19]. 

IDO function in the immune system appears to diverge depending on the type of immune cells and on the origin of the stimulus. Signals such as IL6, IL12, and CD40 ligands favor effector processes and immunity, while signals such as CTLA-4, Foxo3a, and IL10 favor suppression and tolerance. IDO activity gradually increases with the severity of sepsis and is correlated with an unfavorable outcome, but no difference was found between types of infection agent (Gram-positive vs. Gram-negative) or between sites of infection [20,21]. Work by various authors shows that the concentration of circulating Kyn in sepsis significantly exceeds that induced by trauma or hypoxia in SIRS. Increased plasma Kyn values and the Kyn/Trp ratio are early indicators for the development of sepsis following major trauma [19,22].

In critically ill patients, apart from IDO, TDO is also involved in activated Trp catabolism. In contrast to IDO, TDO is not activated by pro-inflammatory cytokines, but its activity is up-regulated by stress-induced release of glucocorticoids, leading to increased amino acid uptake and augmented Trp catabolism. Moreover, long lasting severe depletion of Trp in sepsis patients can have undesirable effects. The activated Kyn pathway leads to the production of nicotinamide adenine dinucleotide (NAD+). With severe Trp deficiency (Trp deficiency), the absence of NAD+ at the cellular level manifests itself as hypoxic symptoms (even with sufficient oxygen) and leads to the development of MODS (multiple-organ dysfunction syndrome) [23].

Blocking IDO is a potential therapeutic strategy for treating septic shock. It improves survival by attenuating the cytokine storm, a major problem e.g., in *SARS-Cov2* [22,23]. When indicated, extracorporeal immunoadsorption by removing pro-inflammatory cytokines will increase both Trp and serotonin (5-HT) levels within 5 days of treatment [24]. Corticosteroids with their potent anti-inflammatory properties, can also be employed. The inhibition of IDO (with 1-methyltryptophan, I3P kinase blockers or JAKs) might have anti-inflammatory effects that are similar to those of corticosteroids. Blocking IDO also limits tissue damage during ischemia-reperfusion, injury induced apoptosis, etc. [24].

On the other hand, IDO can confer tolerogenic phenotypes and limit collateral damage (e.g., in tissue allografts, transplanted tissue, autoimmune disorders, and chronic lung infection) [25,26]. 

### 3.4. Aryl Hydrocarbon Receptor

IDO enables the production of Kyn by innate immune cells to modulate the inflammatory response. The IDO-kynurenine-aryl hydrocarbon receptor mechanism is essential for inducing immunotolerance. IDO expression induces a tolerogenic phenotype not only in APCs, but also in non-immune cells, where it can inhibit immune-effector processes in tissues (chronically infected lungs, graft-versus-host disease, maternal–fetal autoimmunity, diabetes etc.) [5].

Inflammation in innate immune cells, induces the expression of aryl-hydrocarbon receptor (AhR). Kynurenine binding to AhR is immunosuppressive [27,28], as it subsequently results in the production of anti-inflammatory cytokines, such as IL10. The Kyn/AhR complex further initiates phosphorylation of IDO, keeping this enzyme active, further potentiating the production of Kyn, IL10, and TGFβ (tumor growth factor β). The resulting cytokines activate the amino acid transporter SLC7A5 on the T cell membrane that enables Kyn uptake. Kyn/AhR signaling is involved in the generation of T reg (from T naive cells). To conclude, the immunological effects of AhR are complex, and depend on different endogenous and exogenous ligands. IDO/Kyn/AhR signaling can modulate the innate immune response to create an anti-inflammatory microenvironment (Figure 4).

Upon infection, toll-like receptors TLR bind LPS, which acts as the trigger for the IDO/Kyn/AhR immunotolerance signaling pathway by activating IDO and inducing the expression of AhR. Dendritic cells (TLR4 in myeloid DCs and TLR9 in plasma cytoid DCs) that are repeatedly stimulated by LPS induce an anti-inflammatory phenotype by producing anti-inflammatory IL10. 

NFκB signaling plays an indispensable role in the cellular response to LPS; it can signal via canonical (RelA/p65) pathway and noncanonical pathway (Rel/p52). Early on upon stimulation of DCs by LPS, the canonical NFκB pathway (TRAF 3, phospho 65) is triggered, leading to the production of proinflammatory cytokines. Upon repeated stimulation (at later stages of the inflammatory response), the non-canonical NFκB pathway (ReIB, NIK) is preferred, and leads to the production of anti-inflammatory cytokines. This highlights the potential role of the NFκB pathway in mediating LPS-induced tolerogenic phenotype in human DCs through IDO [28].

The second metabolic regulatory pathway of immunotolerance is GCN2 activation during Trp depletion when either IDO or TDO is activated. IDO/GCN2 signaling induces the differentiation of naive T cells to Treg. This signaling role is not limited to immune cells as apart from APCs, IDO is also expressed on epithelial, endothelial and tumor cells—GCN2 activity can thus prevent inflammatory tissue damage [5,6]. Although IDO/GCN2 is a major stress signaling pathway of Trp withdrawal and is considered immunosuppressive (inhibiting cytokine production by APCs and lymphocyte effector function), IDO expression enhances IL6 production in response to microbial signals via a GCN2-dependent signaling mechanism. GCN2 is required for LPS-driven activating transcription factor 4 (ATF4) expression and cytokine production [29,30]. While the details of how the stress-dependent pathway regulates cytokine production are not yet known, it seems to be ubiquitous in the differential regulation in septic and sterile inflammatory situations.

The sufficient availability of amino acids, including Trp, also determines the activation of the mammalian target of rapamycin (mTORC1). Although a direct effect of IDO on mTORC1 has not yet been confirmed, IDO/mTORC1 appears to be another pathway for the metabolic control of inflammation [31].

IDO appears to be part of a network of tolerogenic signals, increasing the local catabolism of several essential amino acids, leading to the rapid removal of Trp from the local environment. This Trp deficiency activates GCN2 kinase and mTORC1, leading to the phosphorylation of eIF2α, which contributes to NFκB activation [29]. Some immunosuppressant drugs mimic the effect of amino acid withdrawal by inhibiting GCN2 and blocking effector Th17 cells. Sustained IDO activity suppresses T cell immunity and the slow progression of chronic inflammatory diseases through immunotolerance [6,30].

### 3.5. Summary: The Importance of Tryptophan in Critically Ill Patients in Terms of Infections, Sepsis

Sepsis is a systemic inflammatory response induced by an infectious agent that contributes to the development of multiorgan failure (multiorgan failure) due to an unregulated immune response. Despite several advances in treatment, the mortality rate remains unacceptably high in critically ill patients. Trp metabolism plays an important role in setting up a controlled inflammatory response by operating to balance both the pro- and the anti-inflammatory aspects of the immune response. 

IDO activation, and the subsequent Trp withdrawal followed by Kyn release, generally acts as a suppressor of immune function dampening inflammatory cytokine production, inhibiting adaptive T cell responses and promoting Treg (Foxo3+) cell function. Nevertheless, under circumstances such as endotoxin shock and sepsis, endotoxin induced IDO expression may cause an excessively pro-inflammatory response in DCs. It is thus clear that IDO serves more than one role in the immune system, actively depleting intracellular Trp stocks leading to significantly increased IL6 production, which can be detrimental during septic shock. IDO inhibition may be an effective target for treating endotoxin shock and may have anti-inflammatory effect similar to that of corticosteroids.

## 4. Role of Tryptophan in the Regulation of Skeletal Muscle Mass

### 4.1. Muscle Mass, and Polyneuromyopathy in the Critically Ill

Adequate muscle mass is essential for critically ill patients, and can determine the outcome of illness, duration of artificial pulmonary ventilation, as well as the quality of life. Skeletal muscle is a plastic “organ”, which in lean individuals can constitute up to 40% of body mass and is a significant reservoir of amino acids. Skeletal muscle mass is determined by the balance between protein synthesis and protein degradation. As a result of acute catabolic disease, where catabolic hormones predominate over the anabolic insulin, energy needs are met primarily at the expense of stimulated proteolysis, leading rapidly to muscle wasting. Amino acids are consumed as a substitute energy source, and even branched chain amino acids (BCAA) are unable to stimulate proteosynthesis under these conditions—a state often referred to as anabolic resistance [32].

Up to 80% of critically ill patients develop the critical condition known as polyneuro-myopathy, also referred to as intensive care unit-acquired weakness (ICU-AW). Apart from the catabolic condition itself (sepsis, cancer, chronic heart failure etc.), analgosedation (or even myorelaxation) with lack of movement or immobilization and artificial pulmonary ventilation also contributes to its development. Skeletal muscle also acts as an endocrine organ, facilitating inter-organ crosstalk. Muscles produce a number of myokines [irisin, neurturin, β-aminoisobutyric acid (BAIBA), etc.], and insufficient nutritive blood-flow to muscles, coupled with insulin resistance/anabolic resistance, restricts amino acids from being used to stimulate proteosynthesis, activating the mTORC1 growth signaling cascade [33].

### 4.2. Proteosynthesis, Proteolysis and mTORC1

mTORC plays a central role in regulating the balance between protein synthesis and protein breakdown and thus controls the volume of skeletal muscle. It is an atypical serine–threonine kinase that exists in two isoforms: mTORC1 that is generally associated with cell growth and mTORC2 that promotes cell proliferation and survival. Its function is influenced by the availability of nutrients (mainly amino acids), growth factors (insulin, insulin growth factor (IGF-1)) and energy (ATP) supply. When these are deficient, e.g., when cells are deprived of amino acids, mTORC1 is diffused throughout the cytoplasm. Upon amino acid addition, mTORC1 rapidly translocates to the lysosomal surface where it interacts with its activator Ras homolog enriched in brain (Rheb), and the transmission of this signal appears to be the main stimulatory function of amino acids. Thus, the rapamycin complex represents a central nutrient sensing pathway that activates proteosynthesis when amino acids—especially leucine and its metabolite hydroxymethylbutyrate (HMB)—are abundant. This can also be exploited therapeutically—by enteral nutrition enriched with HMB. On the other hand, in case of amino acid deficiency, where nutritive signals for muscle cells are missing, mTORC1 regulates autophagy by activating lysosomes and protein cleaving enzymes (caspases) inside them, leading to “cell suicide” [34].

Insulin release following nutritional signals (leucine is an important promoter of insulin secretion) activates phospho-inositol 3 kinase (PI3K) and subsequently protein kinase B (AKT). This pathway is also used by Trp, which activates insulin-growth factor 1 (IGF-1). The other possibility is competitive transport of amino acids into the muscle cell via large neutral amino acid transporters (LATs) and their subsequent activation by an insulin-independent pathway by Ras-related GTPase (Rgas), Vps34 (class III PI3-kinase), and Mitogen-activated protein kinase (MAP4K). The activated rapamycin complex subsequently stimulates proteosynthesis by phosphorylating ribosomal S6 kinase (S6K1) and eukaryotic initiation factor 4E (eIF4E), allowing the translation and synthesis of new proteins to begin. Activation of protein kinase (PKR) results in stimulation of the ubiquitin-proteasome pathway, a major proteolytic system. Its increased activation is implicated in muscle loss and the development of sarcopenia in patients with sepsis, cancer, or increased proteolysis due to aging [34,35,36] (Figure 5).

### 4.3. Effect of Exercise

Fortunately, muscle is a rather plastic tissue that can be influenced by exercise. Muscle fibers adapt to exercise (repeated exercise challenges) both structurally and metabolically. Each individual is primarily endowed with different numbers of type IIB red muscle fibers that contain larger glycogen stores and that respond rapidly to mechanical loading by glycogenolysis. They contain higher levels of lactate dehydrogenase and glycerol-3-phosphate. Type I white muscle fibers, with larger fat stores, are equipped primarily with lipases to utilize fatty acids, whose oxidation is slower. Exercise induces changes in gene expression (expression of gene networks) and energetic adaptation in muscle fibers. Resistance exercise leads to the induction of IGF-1 and the suppression of myostatins, allowing muscle buildup. Endurance training leads to metabolic adaptations, improving exercise tolerance by releasing myokines. An even greater effect, especially on the activation of proteosynthesis and subsequent increase in muscle mass, can be achieved by exercise with the simultaneous administration of BCAA (especially leucine and its metabolite HMB) [28,37]. 

### 4.4. Tryptophan, Kynurenine and Sarcopenia

Trp, like other amino acids, has a stimulatory effect on proteosynthesis. It acts directly by stimulating the rapamycin mTORC1 system. Its supplementation in the diet significantly increases IGF-1 in muscle and leads to the stimulation of proteosynthesis by activating the mTORC1/eIF4/p70s6k signaling pathway. In addition to the Trp signaling cascade directly inducing the expression of myogenic factors in C2 C12 myoblasts, a second Trp metabolic pathway is indirectly involved. Serotonin (5HT) stimulates growth hormone (GH) secretion from the pituitary gland, which increases IGF-1 as well as leptin production in the liver. Trp supplementation thus increases muscle mass and reduces adi-pose tissue. Trp is an amino acid commonly found as a component of total parenteral nutrition or amino acid solutions. The accumulation of Kyn has the opposite effect. The level of Trp decreases with age and this decrease, together with the accumulation of its catabolite Kyn, contributes to the development of sarcopenia [38,39].

Trp may be the key essential amino acid associated with sarcopenia. Metabolomic analyses show an association between low Trp and histidine levels and elevated glutamine and proline levels in age-related as well as cancer-related sarcopenia. As Trp is a glucogenic/ketogenic amino acid, a Trp deficient diet reduces glycolysis in muscle, and in turn increases the concentrations of amino acids, such as Pro and Gln, involved in the tricarboxylic acid cycle (TCA cycle). Glycolysis is linked to the mTORC1 pathway by glyceraldehyde-3-phosphate dehydrogenase binding to Rheb. All of this points to a unique role for Trp in the regulation of skeletal muscle mass, where its deficiency may suppress mTORC1 signaling via reduced glycolysis [31,40].

Exercise also activates LAT transporters, allowing more Trp and Kyn to enter the muscle fiber, while at the same time activating kynurenine aminotransferase (KAT), ensuring the transformation of Kyn to kynurenine acid. Muscle fiber contains only a fraction of the enzymatic equipment compared to liver or immune cells. Under normal conditions, 95% of Trp is metabolized by the Kyn pathway in the liver (TDO). TDO function is inhibited under conditions of stress, high levels of corticoids, inflammation, or sepsis, and Kyn production by IDO occurs mainly in immune cells (with feedback immunosuppression). Kynurenine enters the muscle fiber from circulating blood through LATs and is subsequently degraded to KynA by KAT, to 3Hydroxy-kynurenine (3HK), to quinolinic acid (QA) by kynurenine-3-monooxygenase (KMO), to anthranilic acid (AA) by kynureninase (KynU), and finally to nicotinamide adenine dinucleotide (NAD+) [37] (Figure 6). 

### 4.5. Kynurenine Pathway and Inter-Organ Crosstalk

Looking at the signaling functions of Kyn metabolites in inter-organ communication, it is clear why it is necessary to combat sarcopenia in critically ill patients, and to maintain sufficient lean body mass (LBM), or how exercise affects many organ functions. Kynurenine receptors can be found in various tissues, where they affect a range of functions—influencing the CNS, immune functions, and metabolic activity in a tissue-specific manner (see Table 1).

Exercise with KAT activation in the muscle promotes the utilization of Kyn, reducing its level and thus its availability to the CNS; kynurenines can then manifest themselves fully in inter-organ communication. Exercise during convalescence is not only useful in preventing sarcopenia, at the same time it can also elevate mood and cognitive function, while reducing depression. Thus, it affects the symptoms of post intensive care syndrome (PICS), where it helps to reduce chronic low-grade inflammation, which is the basic condition in this syndrome (see Section 6 below).

Regular exercise also alters muscle metabolism, regulating malate–aspartate shuttle (MAS) enzymes. MAS increases the “energy efficiency” of glycogenolysis by reversibly utilizing malate via oxaloacetate and aspartate to move (shuttle) cytosolic NADPH into mitochondria, increasing glycolysis-related ATP production. The post-exercise elevation of KynA, an important myokine, boosts proteosynthesis increasing LBM. At the same time, it also promotes adipose tissue reduction, a phenomenon termed “adipose tissue browning”, where it engages adipose tissue in metabolically futile cycles, with energy lost to heat production. Among other metabolic functions, KynA restores pancreatic β-cell activity and improves insulin resistance, and the inhibition of N-methyl-D-aspartate receptor (NMDAR) improves insulin secretion and glucose tolerance. KynA reduction of inflammation and intestinal hypermotility explains the positive effect of exercise in patients with irritable bowel syndrome. Furthermore, it has a significant cardioprotective effect, and via GPR35 (G-protein coupled receptor) is involved in ischemic preconditioning [37,41]. 

### 4.6. Summary: Importance of Tryptophan in Critically Ill Patients in Terms of Its Effect on Skeletal Muscle Mass 

Diet, exercise, modulation with BCAA, as well as Trp and Kyn can be crucial tools that can help attenuate muscle loss in critically ill and geriatric patients, and to combat sarcopenia as well as polyneuromyopathy in critically ill patients. The Kyn pathway generates metabolites with biological activities that are critical for inter-organ communication. Thus, the functions of Trp–Kyn metabolism extend to virtually the entire body. Understanding how tissues regulate Kyn metabolite levels, how they affect inter-organ communication, and how this pathway can be used in therapy is thus of great importance not only for critically ill patients.

## 5. The Role of Tryptophan and Its Metabolites in the Development of Depression, Delirium

The neurochemical basis of cognitive function is quite complex, involving many neurotransmitters. Two systems that have well-established roles in cognitive function are those releasing glutamate or acetylcholine as their transmitters and the monoamines norepinephrine, dopamine, and serotonin (5-hydroxytryptamine, 5-HT). The essential amino acid Trp and both of its metabolic pathways lead to the production of a number of neurotransmitters, both through the serotonin pathway with production of serotonin and melatonin, and the Kyn pathway, where quinolinic acid (QA) and kynurenic acid (KynA) significantly influence brain neurochemistry and behavior.

### 5.1. Delirium

Delirium is characterized by a disturbance of consciousness with accompanying alterations in cognition. It is a common complication in critically ill patients (incidence ranging between 30–80% depending on disease severity, and the effects of analgosedation and artificial pulmonary ventilation) [42]. Delirium is a manifestation of acute brain failure, as the brain is one of the first organs to fail in multiorgan dysfunction syndrome (MODS). Overloading the brain’s adaptive mechanisms during acute illness of various etiologies causes a qualitative disturbance of consciousness. Although there are numerous causes of delirium, the two most important ones often occur in the same patient—the action of proinflammatory cytokines and neurotransmitter imbalance. In most cases, the development of delirium is a precursor of a still unrecognized complication, one that should be investigated immediately, and adequate treatment initiated, as delirium significantly worsens patient outcome. In critically ill patients, delirium is most often caused by one or more of a series of complications, such as septic encephalopathy, proinflammatory cytokine-induced postoperative delirium, oxygen free radicals (ROS), development of organ (liver, kidney) insufficiency, adverse drug interactions, or withdrawal syndromes (nicotine, opioids, alcohol) [43].

### 5.2. The Tryptophan—Kynurenine Pathway

TDO in the liver is activated following the stress of acute illness (trauma, burns, surgery etc.) and the release of glucocorticoids, that leads to maximal stimulation of Trp metabolism (whose levels decrease) to form kynurenines. The action of KMO produces 3-hydroxykynurenine (3HK), then 3-hydroxyanthranilic acid (3HAA) through the action of kynureninase, and 3HAA oxygenase catalyzes the formation of the final metabolite quinolinic acid. Kynurenine also undergoes transamination by kynurenine aminotransferase to form kynurenic acid (KynA). Following infection, IDO is activated in monocytes and macrophages (by the action of interferons, pro-inflammatory cytokines), leading to the same metabolic cascade (See Figure 1).

Of the Kyn metabolites, primary neuronal functions are affected by QA (neuroexcitatory) and KynA (neuroprotective role), both acting through N-methyl-D-aspartate receptors (NMDA) of the hippocampus. QA is an NMDA receptor agonist, while KynA is an antagonist of NMDA receptors as well as of α7-nicotinic receptors (α7NR). Under physiological conditions, a balance is maintained between QA and KynA. Under septic conditions, the balance in the CNS is tipped in favor of excitatory QA. Numerous neurotoxic metabolites are produced in activated CNS microglia. Increasing cerebral Kyn levels and decreasing Trp levels produce behavioral and cognitive changes similar to those encountered in psychiatric conditions. In schizophrenia, KynA reduces glutamate release and increases dopamine release, blocks α7-nicotinic receptors, while the neurotoxic and excitatory effects of QA acting through NMDA receptors are seen in dementia. QA initiates and promotes the development of local inflammation in the CNS, synaptic dysfunction, and cell death, which leads subsequently to brain damage [44] (Figure 7).

Aging and acute diseases are both associated with the activation of the immune system. Inflammation, sepsis as well as stress can activate IDO, TDO and the Kyn pathway of Trp metabolism. An increased Kyn-to-Trp ratio (KTR) is typically observed in patients with cognitive dysfunction. Therefore, it is not surprising that the incidence of delirium is more frequent among geriatric and critically ill patients. One must take into consideration that such environmental factors are capable of triggering TDO and IDO in order to prevent so-called “sick behavior” [45]. Apart from influencing cognitive function via NMDA receptors, a KynA effect on nicotinic receptors (α7NR) and subsequent acetylcholine release has been reported. The influence of nicotinic α7NRs on the vagus nerve explains the typical association of sympathetic dominance (hypertension, tachycardia, sweating) in delirious patients. The septo-hippocampal projection to the neocortex seems to be important in aspects of logical thought and learning [44,46].

### 5.3. Serotonin, Melatonin

Activation of Trp metabolism by inflammation and stress through the Kyn pathway leads to Trp depletion and inhibition of the metabolic serotonin pathway. This is initiated by hydroxylation of Trp to oxitryptan by the brain-expressed enzyme TPH2. It is essential for the production of the neurotransmitter serotonin, which is formed when oxitryptan is decarboxylated, and for the pulsatile release of the sleep hormone melatonin, that results upon acetylation and methylation of serotonin (See Figure 1).

Up to 90% of serotonin is produced by THP1 in peripheral tissues. 5-HT is stored in the enterochromaffin cells of the gastrointestinal tract, where it regulates peristalsis, and only 1–2% of serotonin (also referred to as the “happiness hormone”) is produced in the CNS, where it modulates mood, cognitive function, and learning ability. Serotonin reuptake inhibitors function as effective antidepressants [47].

Melatonin is primarily secreted from the pineal gland during the dark period of the circadian cycle. Light information is conveyed by the optic nerve to the suprachiasmatic nucleus (SCN) of the ventral hypothalamus. Since melatonin is also produced in human lymphocytes, it may also have a role to play in the regulation of immune responses. Melatonin has also been shown to possess antiviral properties even against *SARS-Cov2*—something that is currently highly relevant [48,49,50].

Another pathway of indolamine metabolism decarboxylates a minor portion of Trp forming tryptamine, that is then methylated to yield N,N′-dimethyltryptamine, which has a powerful hallucinogenic effect. This dimethyltryptamine likely plays an excitatory role in the development of hyperactive delirium [51]. 

### 5.4. Biomarkers of Delirium

The pathophysiology of delirium is still somewhat unclear. This is not surprising, as it is a highly intricate and complex process that manifests to different degrees under different conditions. We find differences between medical and surgical patients, as well as between the different motor subtypes of delirium (hypoactive, hyperactive or mixed delirium). In this respect, critically ill patients are particularly problematic, given the complexity of neuropathophysiological events. There is no doubt that inflammation and stress are critical in the development of postoperative delirium. Although activation of proinflammatory cytokines can lead to a breakdown of the blood–brain barrier, general markers of inflammation such as CRP, IL-6, TNF, and other cytokines cannot be used in diagnosis. Infection, age, comorbidities with other systemic diseases and many other factors also play a role in determining their levels. Moreover, cortisol levels have not shown positive correlation with delirium in all studies. 

Disturbances in the neurotransmitter pathway (co-)contribute to the development of delirious states. In addition to cholinergic deficiency, which could plausibly lead to delirium and dementia, serotonin, dopamine, and melatonin play important roles in effective cognition, memory, learning, attention, and sleep-wake cycles. Trp, phenylalanine and tyrosine are important as precursors of serotonin, dopamine, and noradrenaline. However, a recent meta-analysis of biomarkers of delirium has yielded only inconsistent results for using amino acids as biomarkers. Among the three amino acids, Trp was the most common, featuring in six studies. While two studies showed an association between Trp elevation and postoperative delirium, three other studies showed an association with Trp depletion and delirium, and the last study with orthopedic surgery patients showed no association. Nonetheless, four other studies have shown an association between low Trp/LNAA (large neutral amino acids) ratios. Three studies observed an association of both reduced and increased levels of melatonin with delirium; but the studies did not discriminate among the subtypes of delirium. Thus, current evidence supporting the use of any one biomarker is clearly insufficient [52].

### 5.5. Summary: The Role of Tryptophan, Its Metabolites in Cognitive Dysfunction, Delirium in Critically Ill Patients

In critically ill patients the brain is exposed to a number of risk factors, and as such, is highly susceptible to the development of delirium. The neuropathophysiology of delirium is multifactorial, although it is clear that the essential amino acid Trp and its metabolic pathways, which are affected by inflammation and stress, are significantly involved in the development of delirium.

The Kyn pathway of Trp metabolism, induced by immunological activation (IDO) and stress (TDO), generates several neuroactive compounds (QA, KynA) and can thus uniquely modulate several cognitive disorders, affecting the development of delirium in critically ill patients. Trp degradation attenuates the production of serotonin and melatonin. Subsequent disturbances in circadian rhythm are important for immune and metabolic functions as well as sleep defects. Critically ill patients often suffer from sleep deprivation with disruption of sleep architecture, which also contributes to the development of delirium. 

This opens up a wide range of possibilities for targeting the pathway for new drug development, with the objective of restoring Trp homeostasis, and thereby influencing the development of delirium. In particular, KDO and IDO inhibitors are currently highly interesting for their potential in reducing the excitotoxic properties of QA and by blocking IDO, in order to restore immune surveillance. KAT inhibitors reduce KynA production and thus dopaminergic overactivity. And lastly, melatonin has a broad spectrum of action such as sleep-initiation, antioxidant, anti-inflammatory and anti-excitatory effects.

## 6. Chronic Inflammation, Post Intensive Care Syndrome

Inflammation has a vital role in the body, where innate immune cells operate to restore homeostasis after injury or infection by eliminating damaged cells. As we age, however, the ability of the immune system to perform these functions gradually deteriorates. These changes are paralleled by the development of a chronic inflammatory state known as inflammaging, which is a term used to describe age-related chronic subclinical systemic inflammation.

Degradation of Trp is part of the innate immune host defense mechanism against infection, it is also involved in a number of other immune functions—in suppressing immunity against chronic infections, suppressing local tissue inflammation and autoimmunity, inhibiting immunity to cancer, participating in tolerance to transplanted organs and maternal tolerance to semi-allogeneic fetal tissue [6]. IDO stimulation under a range of non-infectious stress situations and metabolic stress also predisposes Trp to control inflammaging. Trp is metabolized by the Kyn pathway in order to control age-related inflammation. The Kyn/Trp ratio is a marker of immune activation and can be useful as a biomarker for inflammaging. Trp degradation is an indicator of biological age, and overeating contributes to a decline in the ability of the innate immune system to establish homeostasis and long-term immune tolerance. Inflammaging is involved in the pathophysiology of a number of cardiovascular diseases, type 2 diabetes, and neurodegenerative (Alzheimer’s, dementia) or oncological diseases. Obesity is also an important risk factor, indicating a cross-talk between metabolic and immune functions. Trp deficiency activates the GCN2 signaling pathway reducing immune function and energetic NAD+ production, while at the same time inhibiting the anabolic mTOR pathway [5].

Changes in the composition of gut microbiota accompanied by increased Trp catabolism and indole production also play a role in age-related diseases. In healthy individuals, the resident gut microbiota actively participates in dietary Trp metabolism. It is a source of 5-hydroxytryptamine (5-HTP) for serotonin (5-HT) production in enterochromaffin cells. Serotonin is involved in the stimulation of gastrointestinal motility by acting on neurons of the enteric-nervous system, allowing interorgan crosstalk (primarily the gut-brain axis) [47]. Trp is transformed by bacterial decarboxylase and bacterial tryptophanase into several metabolites, such as tryptamine and indole derivatives (indole-3-aldehyde, indole-3-acetic acid, indole-3-propionic acid, indole-3-acetaldehyde), respectively. A portion of these indoles bind to AhR found in intestinal immune cells. This binding is involved in the maintenance of immune homeostasis at the gut level, resistance to pathogens, and preservation of intestinal integrity. Another portion of indoles (indole-3-propionic acid, indole-3-acrylic acid) binds to the pregnane X receptor (PXr), which maintains the barrier function of the intestinal epithelium and aids mucosal homeostasis. For a critically ill patient, the gut can represent the “engine of sepsis”, and so it is essential to maintain the epithelial barrier (tight junction), gastrointestinal motility and gut microbial composition, which prevent overgrowth of nosocomial bacterial strains [53,54,55].

Intensive care medicine has greatly improved over the last couple of decades, resulting in the improvement of short-term outcomes (28-day survival). However, the long-term outcomes, i.e., the quality of life of survivors, has remained virtually the same. This is in part due to the growing number of critically ill elderly patients. These improved treatments may trigger post-intensive care syndrome (PICS) followed by prolonged illness and poor long-term prognosis. PICS manifests in three areas: cognitive decline (including delirium or dementia), physical impairment (catabolism in acute illness accompanied by immobility is followed by protein breakdown, loss of muscle mass and strength), and mental impairment (depression, anxiety or post-traumatic stress disorder) [55,56].

Clinically, the essential amino acid Trp and its metabolites are involved in all areas of PICS (see Table 2).

## 7. Conclusions

The essential amino acid Trp, with its plethora of metabolites, has an indispensable role in a wide range of physiological functions important for maintaining human health. Trp is essential for the synthesis of proteins, biogenic amines, and energy metabolism.

In critically ill patients, the Kyn pathway of Trp metabolism is activated due to inflammation and stress, further skewing immune balance. Kynurenines are important mediators of inter-organ communication. They are involved in immune response, inflammation, and excitatory neurotransmission. From a clinical standpoint, decreases in plasma Trp and an increased Kyn/Trp ratio are important predictors of an unfavorable clinical outcome. The modulation of Trp–Kyn metabolism using lifestyle (diets, BCAA, aerobic exercise), gut microbiome composition (probiotic based-therapies) and pharmacological interventions could help prevent and treat age-related disease with low grade chronic inflammation and immunosenescence, including PICS, of which persistent low-grade inflammation is a major component. Early mobilization and exercise in critically ill patients amplify skeletal muscle expression of KAT, and shifts peripheral Kyn metabolism towards production of KynA, which does not cross the blood–brain barrier. On the one hand, aerobic exercise can prevent the development of sarcopenia, and on the other hand, can act as an antidepressant as it reduces Kyn levels in the brain.

Promising future directions for pharmacological interventions include targeting the main rate-limiting enzymes, such as indolamine-2,3-dioxygenase (IDO1, IDO2), tryptophan-2,3-dioxygenase (TDO), kynureninase (KYNU), kynurenine monooxygenase (KMO) and kynurenine aminotransferase (KATI-KATIII). Blocking IDO is a potential therapeutic strategy for treating septic shock by attenuating the cytokine storm. Further developments in neurobiology are needed to fully understand the neuroactivity of tryptophan metabolites. The imbalance between KynA (produced by astrocytes) and QA (produced by microglia) may be essential in many diseases, such as delirium, depression, and neurodegeneration. KynA modulates cognition, mood and behavior, and is neuroprotective as it blocks several receptors such as N-methyl-D-aspartate receptor (NMDAR), α7-nicotinic receptor (α7nACh) and α-amino-3-hydroxy-5-methyl-4-isoxazolepropionic acid receptor (AMPAR); QA has the opposite effect, inducing excitotoxicity through these receptors and promoting neurodegeneration. IDO and TDO inhibitors appear to be potentially viable therapeutically approaches to prevent the development of delirium, to slow cognitive impairment, and reduce depression and the accumulation of amyloid-forming proteins.

In summary, Trp modulation may significantly improve the prognosis of critically ill patients. Trp feeding (along with adjusting the Kyn/Trp ratio) can extend lifespan and thus, we can conclude Trp to be a sort of natural medicine.

## Figures and Tables

**Figure 1 ijms-22-11714-f001:**
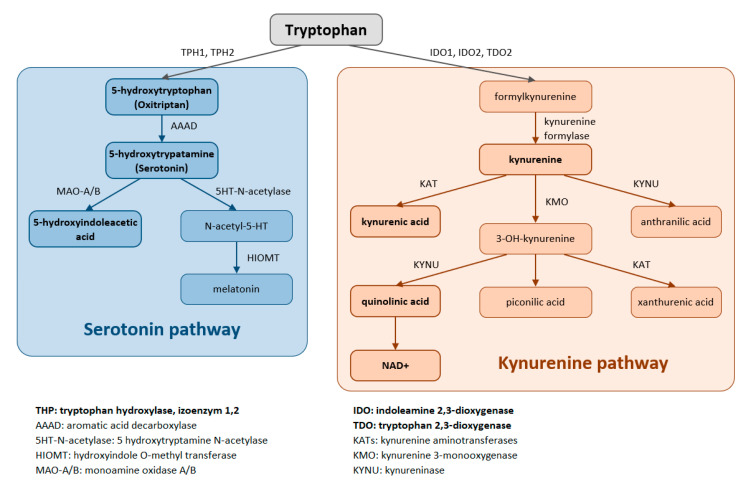
Tryptophan is metabolized by two pathways—the serotonin and the kynurenine pathways.

**Figure 2 ijms-22-11714-f002:**
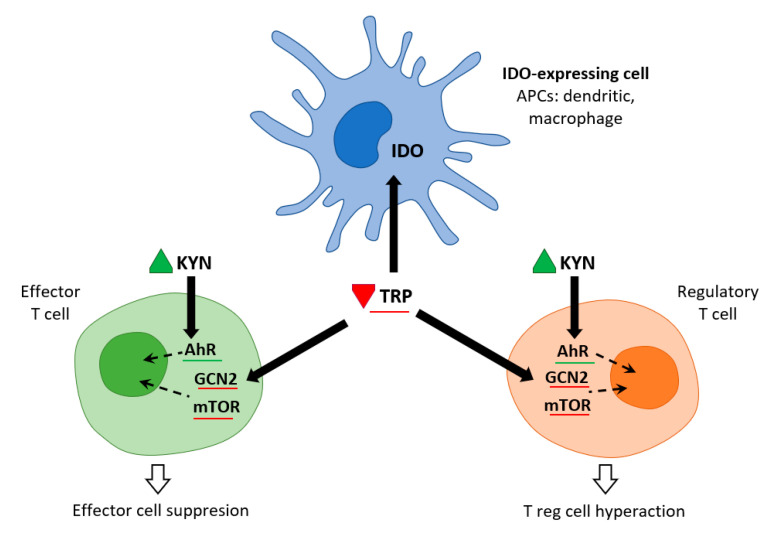
IDO and metabolic control of inflammation (adapted from [6]). IDO activity in APCs leads to Kyn release and Trp consumption. These metabolic signals affect both effector T cells and Treg cells via AhR (Kyn release) and the amino-acid sensors GCN2 and mTOR (Trp consumption), which in turn leads to immune suppression and tolerance.

**Figure 3 ijms-22-11714-f003:**
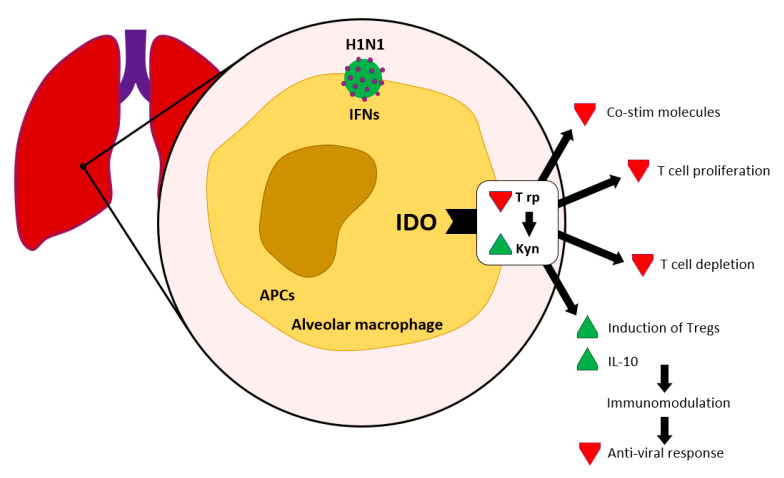
IDO in viral infections according to [11]. Interferon contributes to the activation of IDO, which leads to Kyn production and Trp depletion.

**Figure 4 ijms-22-11714-f004:**
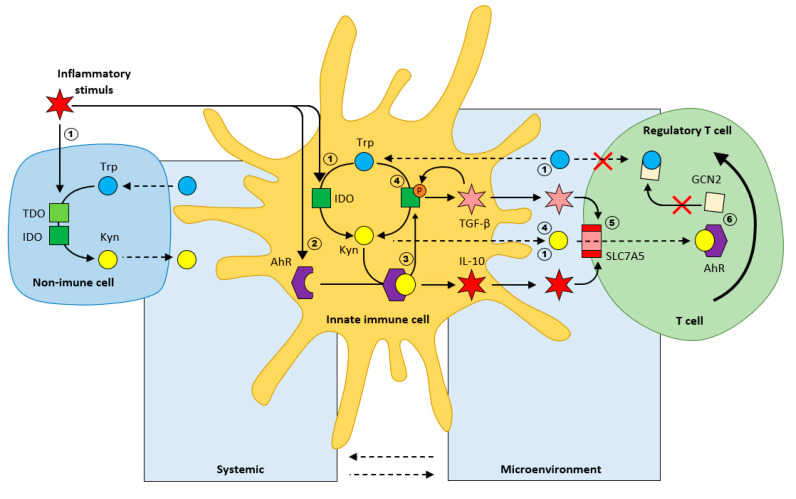
Regulation of inflammation by Trp metabolism (Kyn/Trp ratio) according to [5]. 1. Inflammation activates IDO in both immune, and non-immune cells; 2. Inflammation induces AhR expression; 3. Kyn/AhR results in the secretion of the anti-inflammatory cytokine IL-10; 4. Phosphorylation of IDO results in sustained IDO activity and secretion of the inflammatory cytokine TGF-β; 5. TGF-β induces the amino acid transporter SLC7A5 on the plasma membrane of naive T cells; 6. Differentiation of naive T cells toward regulatory T cells.

**Figure 5 ijms-22-11714-f005:**
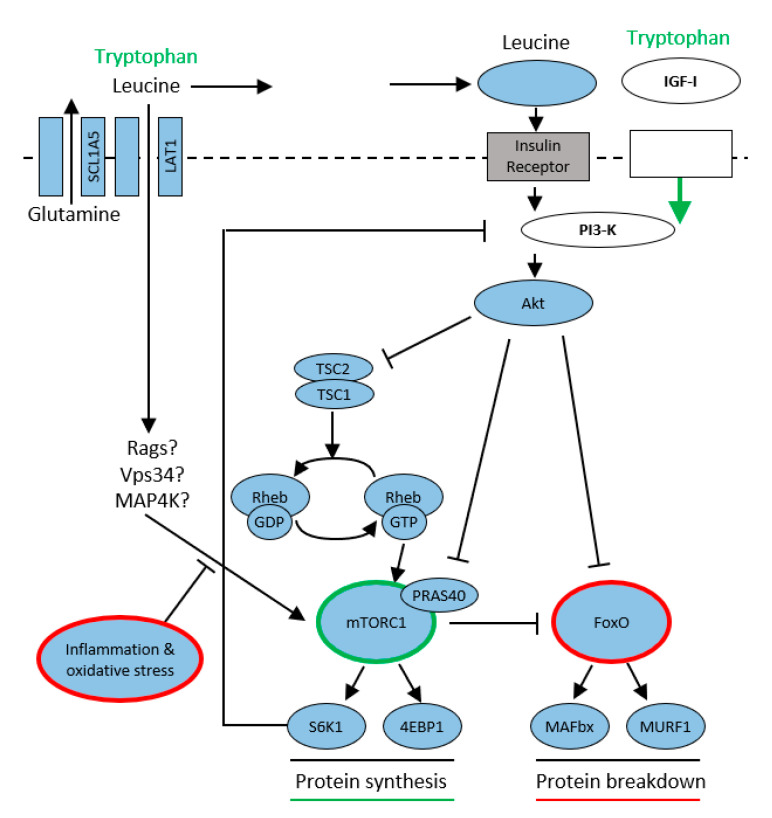
Tryptophan modulates mTORC1/protein synthesis according to [35]. Tryptophan modulates protein synthesis via 2 pathways: (1) IGF-I to PI3-K to Akt to mTORC1; (2) insulin independent mechanism, likely involves Rag, Vps34, MAP4K3. Protein breakdown modulates FoxO. Inflammation and oxidative stress attenuate the activation mTORC1, leading to muscle wasting.

**Figure 6 ijms-22-11714-f006:**
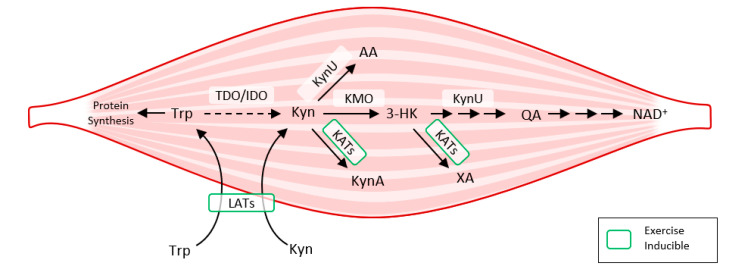
How exercise modulates tryptophan metabolism in skeletal muscle according to [38]. The kynurenine pathway of tryptophan degradation in skeletal muscle. Trp metabolites: Trp, Tryptophan; Kyn, Kynurenine; 3-HK, 3-Hydroxy-kynurenine; KynA, Kynurenic acid; AA, Anthranilic acid; QA, Quinolinic acid; Enzymes: TDO, Tryptophan 2,3-dioxygenase; IDO, Indoleamine 2,3-dioxygenase; KynU, Kynureninase; KATs, Kynurenine aminotransferases; KMO, Kynurenine 3-monooxygenase; LATs, large neutral amino acid transporters.

**Figure 7 ijms-22-11714-f007:**
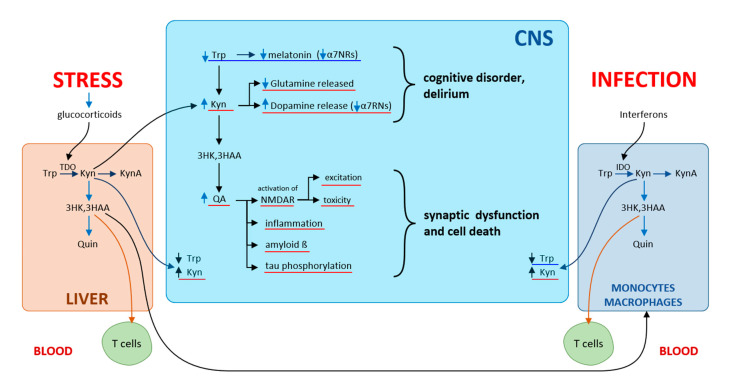
Kynurenines and cognitive disorders according to [44]. The influence of stress on kynurenine pathway via glucocorticoid secretion induces TDO in the liver, while infections via interferons induce IDO in several subsets of leucocytes. Depletion of Trp and increasing cerebral levels of kynurenines produce behavioral and cognitive changes.

**Table 1 ijms-22-11714-t001:** Potential biological activities of circulating kynurenic acid in peripheral tissue/receptors.

Receptors/Coactivators	Organs/Tissue	Functions of KynA
GPR35 PGC 1α1	Skeletal muscle	Myokine ↑ LBM
GPR35 PGC 1α1	Adipose tissue	↓ Adiposity Adipocyte browning ↑ energy expenditure
NMDAR antagonist	Bone	Bone cell maturation Calcium and phosphorus regulation
AhR GPR35	Immune tissue	↓ Inflammation Immune cell communication
NMDAR antagonist	Pancreas	Restore β cell function Improve glucose tolerance
GPR35	Heart	Ischemic protection

**Table 2 ijms-22-11714-t002:** Our hypothesis of the effect of tryptophan metabolism on the development of PICS.

Post Intensive Care Syndrome (PICS) Clinical Signs		Tryptophan Metabolism
Persistent inflammation Immunosuppression	→ ←	IDO activation Kyn increase
Delirium Depression Post-traumatic stress disorder (PTSD)	←	Trp decrease decreased serotonin, melatonin
Physical impairments Polyneuromyopathy	←	Kyn increase, inhibition of mTORC **Persistent Inflammatory Catabolic Syndrome (PICS)**

## Data Availability

Not applicable.

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
