# Peer review of "Tryptophan: A Unique Role in the Critically Ill"

_ijms, 2021, doi:10.3390/ijms222111714_

Round 1

Reviewer 1 Report

In the manuscript, the authors focus on the unique role of tryptophan metabolites in critically ill patients. The topic covered is interesting and additionally very attractive in light of the pandemic situation worldwide. However, the manuscript needs some revisions.

My comments:

  1. Why did the authors omit the role of TDO1? According to the literature data, IDO and TDO1 are highly conserved TDO2 orthologists (Schwarcz R, Stone TW. The kynurenine pathway and the brain: Challenges, controversies and promises. Neuropharmacology. 2017;112(Pt B):237-247. doi:10.1016/j.neuropharm.2016.08.003).
  2. For the reader to better understand the involvement of the kynurenine pathway during sepsis, for example, it is worth presenting all the possibilities of stimulating the activity of IDO and TDO at the beginning. The current description is very cursory. Furthermore, the authors omitted the regulation of TDO activity by tryptophan.
  3. What is the bactericidal mechanism of Trp metabolites such as Kyn, 3HK?
  4. A "Future directions" style subsection in the manuscript is missing. After discussing the involvement of the kynurenine pathway, it is worthwhile to present opportunities for pharmacological modulation or to point out diagnostically relevant parameters.

Author Response

Response to Reviewer 1 Comments

We would like to thank the reviewer for her/his incisive comments that helped us greatly in improving the manuscript.

Please find below our responses to the specific points raised in the review:

  1. Why did the authors omit the role of TDO1? According to the literature data, IDO and TDO1 are highly conserved TDO2 orthologists (Schwarcz R, Stone TW. The kynurenine pathway and the brain: Challenges, controversies and promises. Neuropharmacology. 2017;112(Pt B):237-247. doi:10.1016/j.neuropharm.2016.08.003).

The reviewer is correct. Nonetheless, in much of the literature describing the role of TDO, its effects are not explicitly attributed to the specific TDO1 and TDO2 isoforms. The same is true for the entire body of work on the therapeutic use of tryptophan metabolism [targeting the major enzymes: IDO1, IDO2, TDO, kynurenine aminotransferases (KA- KATI-KATIII), kynurenine monooxygenase (KMO)] in cancer, neurodegeneration, depression, etc. Clinical studies have been carried out on IDO1 inhibitors, IDO2 inhibitors and either TDO inhibitors (680C91, LM10, indazoles, etc.) or dual IDO1-TDO inhibitors.

In the paper mentioned in the comment Schwarcz and Stone write, “loss of TDO2 activity reduces neuronal death. One of the many questions to be addressed is whether similarly impressive effects can be achieved by down-regulation of IDO and TDO1, which are highly conserved orthologs of TDO2”. The authors themselves state that their review focuses on areas of kynurenine research, which are either controversial, or just beginning to receive focused attention.

As intensive-care specialists, we focused on the role of tryptophan and its metabolites from the perspective of the critically ill patient. In this review we attempt to highlight its role in acute and life-threatening conditions such as sepsis, delirium and sarcopenia. The kynurenine pathway has a clear role here. We do not consider ourselves well-enough qualified to answer the question raised by the reviewer and the authors of the paper mentioned. As with IDO1 and IDO2, TDO isoforms presumably have specific patterns of expression in different organs (enzyme expression has been recorded in liver, bone marrow, the immune system, muscle, GIT, urinary bladder, brain - glioma cells, neurons). We have added some text to highlight the organ-specific expression of each enzyme.

We agree that the question raised by the reviewer is pertinent and interesting, and we also hope that additional physiological and pathological roles for brain kynurenines will soon be discovered, and the potential approaches to influence them therapeutically will certainly follow from these discoveries.

  1. For the reader to better understand the involvement of the kynurenine pathway during sepsis, for example, it is worth presenting all the possibilities of stimulating the activity of IDO and TDO at the beginning. The current description is very cursory. Furthermore, the authors omitted the regulation of TDO activity by tryptophan.

We agree with the comment. We want to emphasize the role of tryptophan and especially the kynurenine pathway, and it is important to emphasize, right at the outset, the mechanisms by which IDO and TDO are regulated. We have added some text and highlighted the regulation of the kynurenine pathway between TDO and IDO, and to emphasize the influence of diet and corticosteroids in hepatic control of tryptophan kynurenine metabolism.

In the interests of better readability, we added to the introduction a fuller description of the entire Trp pathway from diet through the gut into systemic circulation, as well as the influence of the microbiota affecting the source of systemic Trp

Citation added: Cervenka, I.; Agudelo, L.Z.; Ruas, J.L. Kynurenines: Tryptophan's metabolites in exercise, inflammation, and mental health. Science. 2017, 357, eaaf9794.

  1. What is the bactericidal mechanism of Trp metabolites such as Kyn, 3HK?

Thank you for your pointing this out, we agree that the effect is more bacteriostatic and not bactericidal. We have modified the text as follows:

… an antibacterial effect, directly suppressing pathogen replication (e.g. toxoplasmosis, chlamydial infection) or limiting the spreading of viral infection

  1. "Future directions" style subsection in the manuscript is missing. After discussing the involvement of the kynurenine pathway, it is worthwhile to present opportunities for pharmacological modulation or to point out diagnostically relevant parameters.

We agree with the comment.

We attempt to summarize each section by mentioning the possible use of tryptophan-kynurenine metabolism modulation via lifestyle (diet, exercise), and pharmacological intervention to prevent and treat sepsis, sarcopenia, and delirium.

We added a paragraph to the Conclusions outlining future directions at the end of the text, especially from the critically ill patient's perspective with regard to the possible impact on PICS (post intensive care syndrome), where sarcopenia, delirium, post-traumatic stress disorders can severely impact quality of life in patients who survive critical illness. We elaborated on the positive influence of aerobic exercise and its effect both on sarcopenia and on the suppression of depression through modulation of Trp metabolism. We also point out kynurenine pathway enzymes, which represent potential therapeutic targets from the perspective of the critically ill with regard to their effect on the development of sepsis, delirium, PTSD. Given the specific focus of our review, we do not elaborate on their role in cancer, schizophrenia and neurodegenerative disorders.

We believe that we have been able, with your insights, to put together a high-quality paper which offers an interesting overview of this topic that is relevant to both healthy people as well as for critically patients.

Sincerely,

Marcela Káňová

Reviewer 2 Report

In this manuscript, the authors address the tryptophan (Trp) and its metabolites from the perspective of the critically ill patient. The authors carefully explain the importance of Trp in inflammation, immunity, and mental health. The term "elixir"  in conclusion is somewhat of an exaggeration, but the argument is not misdirected. This is a well written, interesting, and useful contribution, which I think is entirely suitable for publication in Journal of IJMS.

Author Response

Response to Reviewer 2 Comments

We would like to thank you for your comment. We really appreciate it.

We believe that we have been able, with your insights, to put together a high-quality paper which offers an interesting overview of this topic that is relevant to both healthy people as well as for critically patients.

Sincerely,

Marcela Káňová, Pavel Kohout
